# Machine learning guided aptamer refinement and discovery

Ali Bashir[1,3], Qin Yang [2,3], Jinpeng Wang[2], Stephan Hoyer [1], Wenchuan Chou[2], Cory McLean[1], Geoff Davis[1], Qiang Gong[2], Zan Armstrong [1], Junghoon Jang[2], Hui Kang[2], Annalisa Pawlosky[1], Alexander Scott[2], George E. Dahl [1], Marc Berndl [1], Michelle Dimon [1✉] & B. Scott Ferguson [2✉]

Aptamers are single-stranded nucleic acid ligands that bind to target molecules with high affinity and specificity. They are typically discovered by searching large libraries for sequences with desirable binding properties. These libraries, however, are practically constrained to a fraction of the theoretical sequence space. Machine learning provides an opportunity to intelligently navigate this space to identify high-performing aptamers. Here, we propose an approach that employs particle display (PD) to partition a library of aptamers by affinity, and uses such data to train machine learning models to predict affinity in silico. Our model predicted high-affinity DNA aptamers from experimental candidates at a rate 11-fold higher than random perturbation and generated novel, high-affinity aptamers at a greater rate than observed by PD alone. Our approach also facilitated the design of truncated aptamers 70% shorter and with higher binding affinity (1.5 nM) than the best experimental candidate. This work demonstrates how combining machine learning and physical approaches can be used to expedite the discovery of better diagnostic and therapeutic agents.

---

[1] Google Research, Mountain View, CA, USA. [2] Aptitude Medical Systems Inc., Santa Barbara, CA, USA. [3]These authors contributed equally: Ali Bashir, Qin Yang. ✉email: mdimon@google.com; scott.ferguson@aptitudemedical.com

Aptamers are single-stranded nucleic acid ligands that can be developed to bind a wide range of targets with high affinity and specificity. Comprised of DNA, RNA, or chemically-modified nucleic acids, aptamers offer a unique combination of physicochemical properties that provide important advantages over traditional protein-based scaffolds in certain therapeutic and diagnostic applications. As a therapeutic modality, chemically modified DNA/RNA aptamers are widely appreciated for their non-immunogenic composition and excellent safety profile[1–4]. Their small size and high solubility permit high molar doses and high tissue penetration for maximum bioavailability[5,6]. Their modularity enables easy creation of multi-specific and chimeric agents[7–9], conjugates for extending pharmacokinetics[10] or delivering payloads[11,12], and reverse complement "antidotes"[13,14]. In diagnostics, aptamers offer high stability, facile manufacture and can be engineered into molecular switches to enable continuous monitoring[15,16]. For any application, the quality of an aptamer is determined by its sequence which must be derived from a discovery process.

High-quality aptamers are exceptionally rare occurrences in the sequence space[17]. Discovering them typically entails sampling the space by creating a large random library of candidate nucleic acid sequences[18–22] and enriching for the candidates with desired characteristics. To yield the best possible aptamers one must maximize the effectiveness of the discovery process and the library itself. Most methods have focused on the former, seeking to improve the traditional aptamer discovery process, SELEX, which identifies aptamers through an iterative process of selection[23,24]. While a number of methods have been published to improve aptamer affinity, specificity, and success rate[25–29] all experimental approaches are constrained by the physical number of aptamer candidates in the library and the physical approaches to synthesize them. Practical limitations constrain the library size to ~$10^{15}$ candidates, which covers only one billionth of the sequence space available to a typical 40-base aptamer library, for example[30]. The library is further limited by the means of sampling the sequence space since fully-specified oligo libraries can only reach a diversity of ~$10^6$ sequences[31] and scalable random strategies[32,33] fail to precisely explore the sequence space. Overcoming this limit via intelligent sampling strategies could yield significantly better aptamers.

Although appreciably sampling the sequence space via physical methods is not feasible, in silico approaches offer a compelling opportunity to search the space more efficiently. Chushak and Stone used secondary structure prediction and computational docking to filter the starting sequence pool[34]. Knight et al. used machine learning (ML) to model a sequence-fitness landscape by training a random forest model on aptamers and their corresponding affinities[35], though this model was not used to generate novel sequences. Modern neural network (NN) approaches have had success generating sequences in other biological domains. Predictive models trained on experimental results have been used to successfully computationally evolve yeast 5′UTR sequences with higher protein expression[36,37] and antibody sequences with greater specificity[36,38]. Predictive ML models offer the opportunity to perform directed aptamer design: synthesizing and evaluating sequences in silico to dramatically reduce the number of sequences required to be experimentally screened.

In this work, we develop and validate a ML-guided Particle Display methodology (MLPD) to improve existing experimental candidates, identify completely novel DNA aptamers, and truncate aptamers to improve therapeutic utility. To demonstrate the method, we selected the target protein neutrophil gelatinase-associated lipocalin (NGAL). NGAL is an emerging diagnostic and prognostic biomarker of acute kidney injury and biomarker of urinary tract infection[39,40]. Using results from an initial PD screen on NGAL, we trained machine learning models to predict affinity. We used these machine learning models to predict improvements to experimentally-derived aptamers and to predict aptamers de novo, experimentally evaluating 187,499 aptamers. In addition, we extended our approach to automatically identify candidates for another important practical application: yielding minimal-length aptamers that maintain binding affinity. Taken together, MLPD is able to more efficiently explore the fitness landscape for aptamer design.

## Results

**MLPD: machine learning guided particle display.** MLPD combines state-of-the-art experimental and computational approaches to generate high-affinity aptamers. The process starts with a traditionally synthesized library, which can be viewed as a sample from the space of possible aptamers (Fig. 1a). To produce training data for machine learning models, particle display (PD) was used to measure the relative affinity of every aptamer candidate in the library based on the target concentration and the measured fluorescence (Fig. 1b)[29]. By varying the target concentration to control the stringency, PD was used to partition this library into positive and negative aptamer pools at multiple affinity thresholds, each of which was characterized via Next Generation Sequencing (NGS) on the Illumina NextSeq (see Sequencing and data processing in the Methods section for details). These DNA sequences were passed as input to the ML method. Fully connected and convolutional NN models took these features and predicted affinity measurements (Fig. 1c, described in detail in the Methods subsection; ML models, features, and output layers). We evaluated the models using a 20% test subset of the original PD dataset. In order to assess whether ML can generate useful aptamers, three high-performing models were used to create sequences. Our approach followed a two-step process. First, we identified seed sequences to serve as starting points for ML-guided mutation. Next, we iteratively mutated these seeds for five rounds, selecting mutated sequences in each round that were preferred by the model (Fig. 1d). Aptamers were generated from three sets of initial seeds: (1) high-performing aptamer sequences from the initial PD, (2) random sequences that were screened in silico, and (3) as a baseline, completely random sequences. Seed sets were then iteratively mutated, with each variant being scored with an ML model to enrich for mutations preferred by the model, the output being an in silico enriched pool containing novel candidate sequences (Fig. 1d). Lastly, these candidates were synthesized and experimentally measured via PD (Fig. 1e). In principle, this process can be repeated (Fig. 1b–d) until sufficient high-quality candidates are obtained.

**Particle display partitions the library based on affinity threshold.** We aimed to create ML training data with minimal false positives and multiple, sharply-defined affinity levels (Fig. 2a; Methods). We synthesized aptamer particles from a DNA library containing a 40-mer random region flanked by primer sites and performed two rounds of PD with a total of three affinity thresholds. To perform a round of PD, the pool of aptamer particles was first incubated with fluorescently-labeled target at a given concentration. Next, the pool was screened via fluorescence-activated cell sorting (FACS) to partition the particles based on a gating fluorescence value as the threshold (see Methods for details). The affinity thresholds were separated by fourfold each by lowering the target concentration fourfold while keeping the same sorting gate value at $F_{max}/3$ (Supplementary Table ST1). The approximate affinity thresholds for the increasing stringencies (<128 nM, <512 nM, and <2 μM) were estimated

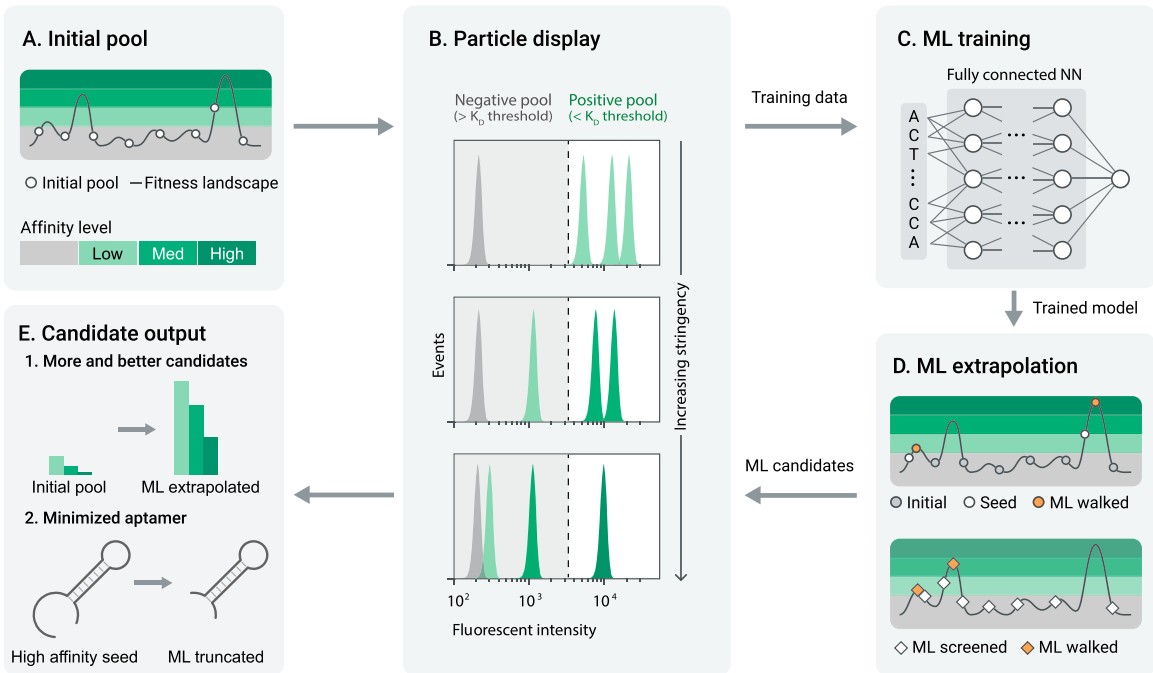

**Fig. 1 MLPD overview. a** Aptamer candidates from a physical pool (white dots) sample a small portion of the fitness landscape (dark gray line), each with a corresponding affinity level (light to dark green). **b** Particle display discerns the affinity level of each candidate by interrogating the library at multiple stringency levels. **c** Aptamer sequences and their corresponding affinity levels are used to train and validate a neural network ML model. **d** The ML model extrapolates new sequences on the fitness landscape in two ways: (1) mutating existing candidates (white dots) in a model-guided fashion (orange dots), and (2) nominating novel sequences in silico, predicting their position on the fitness landscape (white diamonds) and walking top-performing sequences to higher affinity levels (orange diamonds). The extrapolated candidates are synthesized and experimentally tested. **e** MLPD yields more candidates at each affinity level compared to the initial library, and enables sequence truncation without reduction in affinity.

via $K_D$ curves on a subset of the observed aptamers (Supplementary Fig. S1, Supplementary Table ST1). In Round 1 the relatively low-affinity thresholds (<512 nM, and <2 μM) created a large pool of initial candidates. Under these conditions, we observed that a fraction of the library was able to pass the $F_{max}/3$ thresholds (Supplementary Fig. S2). At each concentration, we collected the positive aptamer particles as well as ~$10^5$ negative aptamer particles (F < $F_{max}/3$), for use as negative examples in ML training. Primer particles exhibiting no aptamer sequences were excluded from this analysis.

To enrich our libraries with additional high-affinity binders, we conducted a second round of PD using all three stringencies (Fig. 2a). Two Round 1 positives pools were amplified, then mixed at a 1:1 ratio to serve as template for Round 2 PD. We observed a much higher proportion of positive aptamer particles (1.0–6.5%; Supplementary Fig. S2B). As in Round 1, positive and negative aptamers were collected at each concentration, amplified, and indexed for NGS sequencing. We verified that the particles made from individual pools were enriched at respective affinity thresholds and more than 80% of aptamer particles sorted from round 1 to round 2 had the desired affinity (Supplementary Fig. S2C).

To verify the consistency of the PD, we evaluated the concordance of observed positive sequences at each affinity level. Each aptamer was labeled as successfully passing an affinity level if it satisfied two criteria. First, the aptamer had to be reliably detected in the positive pool, defined as having a sequencing count of at least 20% of the expected bead coverage (corresponding to 1373, 1469, 1311 sequencing counts for the 2 μM, 512 nM, and 128 nM stringencies, respectively; Supplementary Table ST2). Second, the aptamer had to be more prevalent in the positive pool than the negative pool as calculated by the aptamer's normalized sequencing fraction within positive and negative pools. All

positive pool counts, on average, exceeded negative pool counts by >29x for sequences passing a stringency threshold. For both the original PD (Fig. 2b) and the MLPD (Fig. 2c) most sequences observed at a given affinity were observed in all lower affinity levels, indicating high-fidelity partitioning.

**Trained ML models can predict high-affinity aptamers in a held-out dataset.** The NGS-sequenced positive and negative pools served as training data for a neural network (NN) model. For the input, our models used the concatenation of a simple one-hot and kmer count-based representation of the input sequence (see Methods subsection, ML Model Design). Multiple prediction tasks were implemented as described in our Methods, each of which fell into these two general classes: (1) predicting the abundance (sequencing fraction) of the input sequence in the selection pools (Counts model), and (2) treating each PD experiment as quantized measurements indicating whether the aptamer exceeded the affinity level set in FACS. We implemented the second class in two ways. First, the Binned model has a separate output for each stringency level, similar to the Counts model, except instead of predicting sequencing fraction per pool, the model predicts if a sequence will be present at each stringency level. Second, the SuperBin model reduces these multiple output predictions into a single number representing the maximum stringency level where the sequence is predicted to be present. In all three prediction tasks, the model is trained with least-squares regression.

An important consideration for machine learning is to maintain separation between data used to train the models and data used to validate their performance. Since sequencing error can create multiple data points from the same underlying aptamer sequence, it was necessary to cluster sequences to ensure

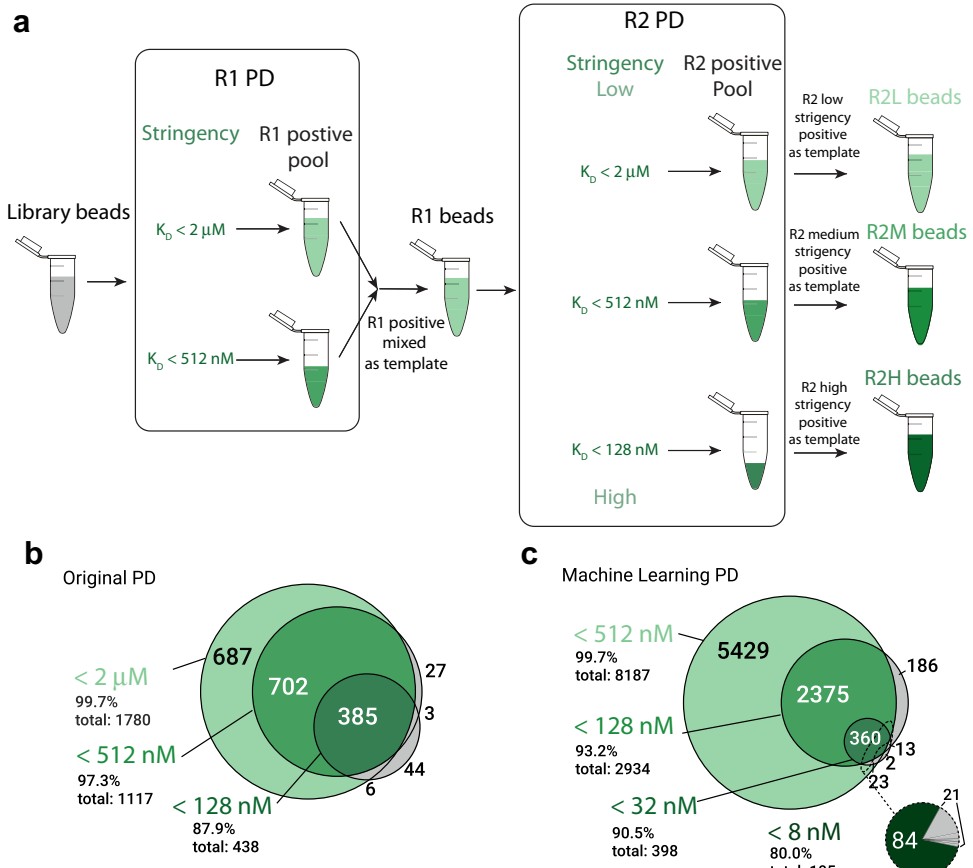

**Fig. 2 Design of particle display training data and concordance across experimental affinity thresholds. a** Two rounds (denoted R1 or R2, respectively) of particle display (PD) experiments were run with increasing stringency (decreasing protein concentrations) such that the lowest stringency (light green) should contain all aptamers observed at higher stringencies. At each stringency level, we obtained positive pools of aptamers each with affinities that pass the affinity threshold (green shades) and negative pools that do not pass the affinity threshold. R1 positive pools were amplified then mixed as the template for the R2 particle display experiment. All pools were NGS sequenced. **b**, **c** Venn diagram of unique aptamer clusters in (**b**) the original particle display experiment and (**c**) the machine learning guided particle display (MLPD) positive pools. Green-colored sections indicate sequences observed at a particular stringency and all lower stringencies. The dotted line and pie chart in (**c**) show the concordance (dark green) of the fourth and highest stringency run in the MLPD experiment (< 8 nM).

these similar sequences did not appear in both training and testing sets. To create a clean train/test split, the initial 500,454,107 reads that passed quality filtering were clustered into 910,441 clusters using a conservative Levenshtein distance of 5 and clusters were never separated across train and test datasets. Machine learning models were trained on 80% of the sequences, and the other 20% were used to evaluate the trained models. Multiple models were trained using Vizier[41], varying hyperparameters, number of fully connected middle layers, and output layer architectures. These were evaluated based on their performance in predicting "positive binders," defined as sequences in the top 1% of the test set. All selected models had three convolutional layers and between one to four fully connected layers (see Methods, Supplementary Fig. S6).

To compare models trained on sequence abundance with those trained on the binned values, we evaluated all the models on their ability to predict the test dataset via two different metrics: the normalized sum of counts from Round 2 < 128 nM and Round 2 < 512 nM, or the highest stringency-level for the subset of aptamers with consistent labels shown in Fig. 2c. In both metrics, the models were better at predicting the very top candidates (Top 1% and < 128 nM) compared to candidates with lower affinity (top 5–10% and < 512 nM-2 μM), suggesting differentiating properties unique to this subset. Despite the Counts model being trained on richer input, it did not show improved performance

relative to the simplified Binned representation. In fact, it had reduced performance on most metrics, reinforcing the intuition that the true signal provided by PD is simply the observation of a sequence passing an experimental affinity threshold.

**Trained ML models can predict novel aptamers with high affinity**. Next, we evaluated the predictive power of our models by generating model-guided sequences and validating them experimentally. Exhaustively simulating the entire fitness landscape is computationally infeasible; instead, we developed a model-guided mutation strategy to "walk" from seed sequences to mutant sequences with higher predicted affinity. Seeds defined a starting position in the fitness landscape. In each model-guided step, seed aptamers were randomly mutated (with 0-2 single nucleotide (nt) substitutions) and scored. The top mutated aptamers were set as new seeds and the process was repeated for five rounds. Aptamers from each round of mutation were selected, synthesized, and tested via PD. In the MLPD design, four stringency levels were run, again increasing by fourfold per level. The estimated affinity thresholds for these stringencies were 512 nM, 128 nM, 32 nM, and 8 nM (Supplementary Table ST3 and Supplementary Fig. S1).

We selected seed sequences in three ways to characterize different aspects of the model. First, a purely random baseline (177 "random seeds") was used to evaluate if the models had

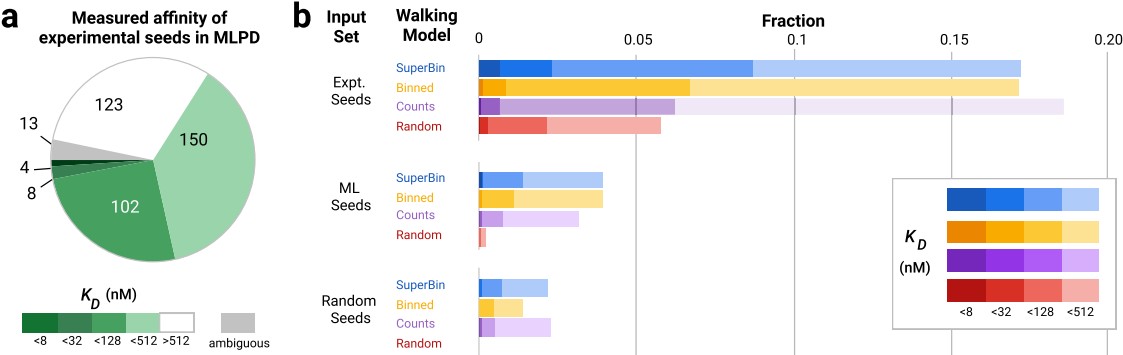

**Fig. 3 Experimental validation of machine learning predictions. a** Observed affinity for experimental seeds used in machine learning guided particle display (MLPD). Greens correspond to sequences at particular affinity thresholds (with darker greens indicating higher affinity); white corresponds to sequences below the lowest screened affinity, and gray corresponds to sequences with ambiguous affinity. **b** Candidates generated by three machine learning (ML) model walks (SuperBin (blue), Binned (orange), Counts (purple)) and Random walks (Red) as a fraction of the input pool size. The MLPD panels show ML-directed walks starting from (top) the original particle display experimental (expt.) seeds, (middle) randomly screened and model ranked ML seeds, (bottom) completely random seeds. Independent of the seed category, the ML-directed walks substantially outperform random walks and the original particle display.

learned general sequence properties that could be applied to arbitrary sequences. Second, a mixture of high performers within the original PD experimental pool (400 "experimental seeds") was used to evaluate the model's ability to enrich for alternative candidates near experimental starting points. Third, top performers through computational model screening of random sequences (14,977 "ML seeds") were used to evaluate whether the models could generate diverse, high-quality candidates de novo. For ML seeds, each model ranked 1 billion random sequences, selecting the top ~5000 (0.0005%). All three seed sets were walked using the Counts, Binned, and SuperBin models as well as a Random walk baseline, yielding a total of 82,931 aptamers (Supplementary Table ST4).

In each seed set, the ML models substantially outperformed random walks (Fig. 3). The experimental seeds, selected from the top aptamers after two rounds of Particle Display, performed well on their own, thus most mutations on these sequences were deleterious, especially for seeds with initial $K_D < 128$ nM (Supplementary Tables ST5 and ST6). Despite this challenge, the ML-directed walks from experimental seeds improved 11.3-fold (8 nM) and 4.6-fold (32 nM) over random walks. Notably, the model performance appeared to be independent of the distance to training set sequences. Model AUCs on randomly walked sequences from experimental seeds in the training set did not decline as the distance increased (Supplementary Table ST7).

Although the ML directed walks from the ML seeds did not yield as many high-affinity sequences as walking from experimental seeds, they markedly outperformed random seeds, and were able to generate candidates with PD affinity levels beyond the training data at the most stringent level tested in MLPD (8 nM). Walked aptamers from ML seeds were quite distinct; all walked sequences were a Levenshtein distance of 10 or more from any experimental seeds, indicating that the ML models have generated de novo sequences, not just improved existing candidates. Across all ML models, we saw a 460-fold increase in the fraction of <128 nM aptamers compared to PD (Fig. 3a), which increased to 1214-fold when using the MLPD recalibrated affinity for top PD candidates (Supplementary Fig. S3). Interestingly, while the SuperBin model did not have the best AUC performance on the test set (Table 1) it produced the highest fraction of sequences at all affinity levels <128 nM across seed sets.

To validate these observations, full $K_D$ curves were generated for a subset of aptamers, including a portion observed in PD and

a portion predicted by ML (Supplementary Table ST2). In addition to verifying the fourfold increase between stringency levels for both PD and MLPD, the $K_D$ curves confirmed the improvements gained via the ML-based walks: in one example improving a seed with an affinity of 275 nM to 8 nM. The highest performing experimental seed was outperformed by several ML-predicted aptamers. Surprisingly, the best performing aptamer candidate was not derived from an experimental seed, but rather from an ML-guided walk of a randomly screened ML seed. This demonstrates that MLPD can yield better aptamers than the examples it was trained with.

**ML Models can identify motifs and subsequences enriched for binding affinity.** Sequence motifs are known to be important in aptamer affinity[42]. We first sought to identify the most frequent motif observed in aptamers selected by the ML models. To eliminate potential biases introduced by non-random seed sequences, we first examined differential enrichment between walked sequences and seed sequences in the "random seed" set. Using MEME[43], we obtained a 7 nt motif (consensus motif = TGGATAG, $e$ value = $3.2 \times 10^{-18}$) shown in Supplementary Fig. S4A. Next, we examined the original particle display test sequences that were not used to train the model. Experimental test sequences observed in a positive pool were compared to the full set of experimental test sequences. This yielded a highly similar 7 nt motif ($e$ value = $3.3 \times 10^{-86}$) that shared the same consensus sequence, TGGATAG (Supplementary Fig. S4B). Interestingly, while the motif was not explicitly discussed, a recent, independent study of high-affinity aptamers for NGAL included three distinct aptamer candidates also containing TGGATAG[44].

A common next step in aptamer generation is minimizing sequence length. Structural studies of aptamers have shown that a full length (~80 nt) aptamer is usually unnecessary to retain strong binding affinity. Minimization, or truncation, can reduce synthesis cost, complexity, and potential for non-specific interaction all while increasing the maximum attainable concentrations (surface or volumetric)[45]. Traditionally this is achieved via brute force experimental approaches[46] or by researchers' biological insight coupled with manual curation to investigate enriched motifs[47] and secondary structures[48,49].

Given that the models appeared capable of learning features important for affinity, we established a simple data-driven

**Table 1 Test set AUC using two different criteria.**

| ML model name | Output targets | Normalized Rd 2 Count Sum AUCs | | | Affinity threshold AUCs | | |
|---|---|---|---|---|---|---|---|
| | | Top 10%: 563 seqs | Top 5%: 281 seqs | Top 1%: 56 seqs | 2 µM: 409 seqs | 512 nM: 215 seqs | 128 nM: 84 seqs |
| Counts | Counts + PF | 0.62 | 0.73 | 0.84 | 0.76 | 0.85 | 0.86 |
| Binned | Stringency Labels | 0.62 | 0.78 | 0.89 | 0.82 | 0.91 | 0.95 |
| SuperBin | Stringency summary value | 0.63 | 0.76 | 0.83 | 0.79 | 0.87 | 0.87 |

Affinity threshold AUCs utilize the three stringency thresholds while Count Sum AUCs calculate the top fraction of sequences observed in Round 2 at the two highest stringencies. All models show improved prediction performance at increasing stringency levels with the binned model performing best at nearly all stringency criteria and levels.

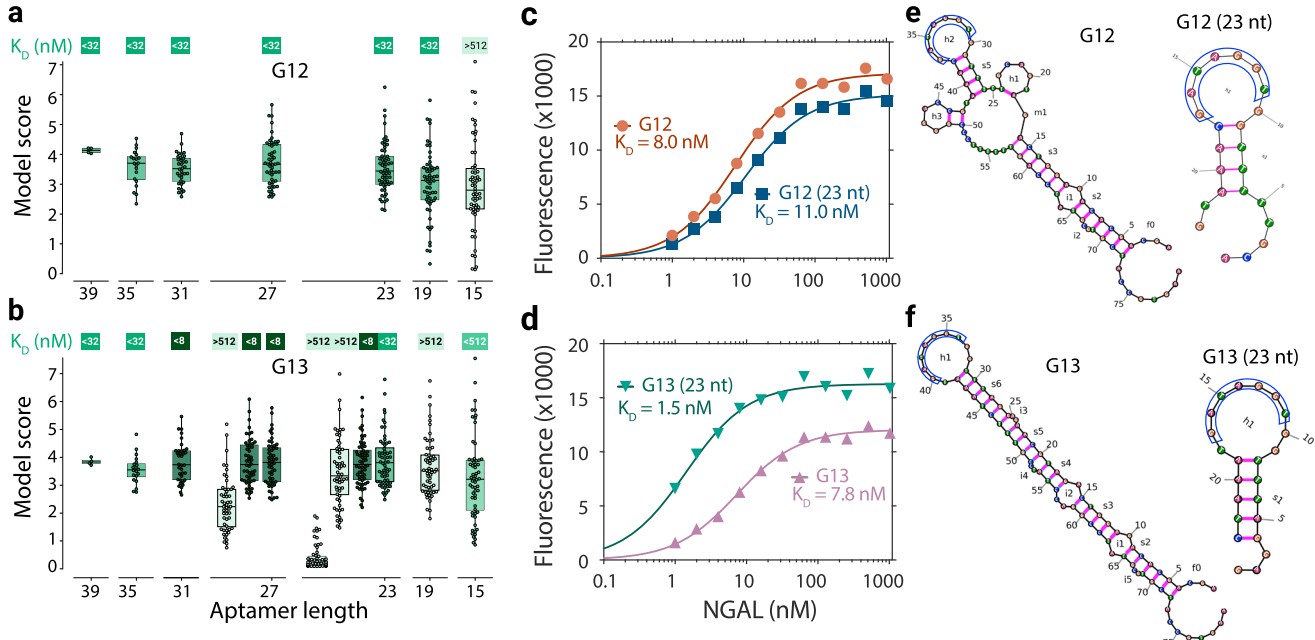

**Fig. 4 Performance of machine learning directed aptamer length truncation. a, b** Box-plots and swarmplots showing model scores for candidate aptamers calculated across multiple sequence backgrounds. Each swarm/box plot corresponds to one core sequence: each point represents the core sequence in a different sequence background, each box represents the median, lower, and upper quartiles, and whiskers correspond to 1.5x the interquartile range. Sequence lengths with multiple swarm/box plots indicate cases where multiple different subsequences of the same length were tested experimentally. Particle display affinity level for a truncated sequence is shown by shade of green in the corresponding swarm. **c, d** $K_D$ curves showing the affinity of the full-length sequences (orange, purple) and 23 nt truncations (blue, teal) for G12 and G13, respectively. **e, f** Secondary structures of the full-length sequence and 23 nucleotide (nt) truncations for G12 and G13, respectively. Each nt is indicated by a small circle (A (maroon), C (blue), G (brown), T (green)). Covalent bonds in the phosphodiester backbone are shown in black and hydrogen bonds between bases are shown in magenta). The TGGATAG motif is outlined in blue.

approach to identify high-affinity core sequences within candidate aptamers. To test smaller sequences using models trained on fixed-length sequences, each substring of length n was placed at all possible positions within the aptamer. Next, the surrounding sequence was filled in with different background sequence (homopolymers of either A, C, G, or T) and scored to get a distribution of affinities for the core subsequence. We performed the experiment from 2 of the <8 nM aptamers, PD derived G12 and ML derived G13. All substrings of 15, 19, 23, 27, 31, 35, and 39 nt were evaluated by the SuperBin model and a subset of these core sequences were experimentally validated. Experimental details for this truncation study are described in the Methods section.

The distribution of model scores was quite wide (Fig. 4, Supplementary Table ST8). While the median model score of core sequences did not always lead to the best candidates, core sequences that consistently produced high model predicted scores

irrespective of their position within the background sequences more frequently maintained the performance of the full-length sequence. This allowed us to identify aptamers as short as 23 nt that met our most stringent affinity condition (Fig. 4b). We experimentally verified these high affinities by measuring the $K_D$ for each full-length sequence along with its top 23 nt candidate (Fig. 4c, d). The G12 truncation yielded a 23 nt candidate with $K_D$ close to that of the full-length sequence (11 nM vs 8 nM). In the case of G13, the 1.5 nM truncated aptamer surprisingly yielded a 5.2-fold improvement over the full-length sequence.

In both cases, the high-performing 23-mer cores (ACGTT TTTGGTGGATAGCAAATG and GAGGATTTGGTGGATA GTAAATC) were contained in all larger truncations that reached the same $K_D$ threshold suggesting these to be the key binding subsequences from the full-length sequence. Secondary structure prediction revealed that the 23 nt truncated aptamer of G12 and G13 shared a hairpin structure with the TGGATAG motif located

on the loop region (Fig. 4e, f). The same structure was observed in the full-length sequence for G12 and G13, suggesting the likely importance of this structure. The high affinities observed were particularly notable given the degree of truncation and performance relative to the original PD data. Given the original aptamers used for training were 80 nt, the 23 nt aptamers represent elimination of >70% of the original sequence length. Furthermore, the full-length G13 sequence was obtained via an ML-directed walk from an ML-inference seed. Therefore, the G13 truncation exhibited 5.3-fold higher affinity than the best-observed candidate from the original PD, despite having never started from any experimentally derived candidate.

## Discussion

ML-guided PD has immediate benefits for researchers seeking to increase the diversity of candidates into therapeutic pipelines or locally optimize existing candidates, and is broadly applicable for general aptamer development. As a 25 kDa soluble protein, the composition of NGAL is not atypical of protein targets. The straightforward domain-agnostic features in the models highlight the strength of a data-driven approach and increases the likelihood that such a model would translate to other targets.

Surprisingly, training on a large volume of relatively low stringency data (<128 nM) enabled extrapolation to candidates with orders of magnitude higher affinity. Ultimately, the value of a machine learning model is in the predictions it makes which, in this work, corresponds to the creation of high-affinity, experimentally validated aptamers. Notably, a limitation of our approach was the use of a single test set (holding out 20% of the data) for hyperparameter tuning. Given the limited positive examples, cross-validation could improve robustness of the tuned models. Despite having a relatively low AUC, the models substantially enriched for new aptamers with high-affinity. In particular, models trained on particle display bins (with comparatively limited positive training examples) seemed to outperform the more fine-grained signal employed in the Counts model. This increases the potential applications of these approaches since obtaining many high-affinity aptamer candidates experimentally is often a major challenge. While promising, there are some natural extensions to the protocol which would increase its utility.

Increasing the number and diversity of positive training examples would yield the most straightforward gains in MLPD. Experimentally, this could be achieved through either larger input libraries or SELEX pre-enrichment[29]. In addition, one could extract more potential binders for model training from each sequencing round by using approaches that incorporate structure to identify low-abundant aptamers with binding affinity[50]. Similarly, one could screen more random sequences in silico. While computationally screening the entire sequence landscape is infeasible, our naive in silico mutagenesis strategy demonstrated that model-guided evolution of candidates can discover desirable aptamers at higher rates than would be expected given the number of model-evaluated sequences. While a more aggressive mutation strategy to optimize model scores may be possible, such approaches may lead to pathological behavior in NN models[51]. The chosen strategy attempted to mitigate this to identify diverse, high-scoring candidates. More sophisticated exploration-exploitation algorithms could dramatically reduce experimental iterations while increasing concordance between the model and ground truth; for example, an active learning approach, where the model is retrained after each round of experimentation.

In addition to efficiency gains, MLPD can move beyond what is possible experimentally to explore multiple parameters critical for therapeutic aptamers. The ability of ML models to simultaneously predict multiple properties could be used in a variety of ways. For example, to discover aptamers on the basis of both affinity and specificity[4], one can perform screens on a desired target and non-targets such as undesired homologues or protein mixtures, as we demonstrated in our previous work, Multi-Parameter Particle Display[28]. By preferentially walking up the affinity landscape for the target and down the affinity landscape for the non-target, one could increase affinity and specificity simultaneously. By walking up both affinity landscapes, one could increase cross-reactivity. Analogously, other screens could be designed to partition aptamers based on additional measurable parameters such as dissociation kinetics or nuclease stability. Due to the combinatorial explosion of conditions, simultaneous optimization of such multi-parameter fitness landscapes is nearly impossible experimentally; even if such screening was feasible, the likelihood of observing an aptamer at the intersection of a desired property set may be quite low. However, by transforming experimental results into N-dimensional property vectors, candidates across all parameters can be optimized in silico.

This work comprises a fundamentally different means to generate high-quality DNA aptamers providing a potential avenue to optimize multiple properties that are desirable for clinical applications. As a complementary approach to purely experimental aptamer selection, our work can be scaled and combined with the wide range of existing aptamer research as well as other directed evolution research currently underway, opening up an exciting area of development. More broadly, we show that the combination of traditional bioanalytic methods with machine learning enables outcomes that were otherwise not possible via experimental means alone, not only accelerating life sciences research but also enabling new research questions to be asked.

## Methods

**DNA aptamer particle synthesis**. The single-stranded DNA library and primers used for the research are synthesized by IDT (Supplemental table ST10). A step-by-step protocol describing the DNA aptamer particle synthesis and particle display screening can be found at Protocol Exchange;[29,52] it is summarized briefly as follows. The first step is to create FP-coated particles. To do so, 5′-amino-modified FP (5′-amino-PEG18-AGCAGCACAGAGGTCAGATG-3′, 0.2 M) was covalently conjugated onto 1-μm MyOne carboxylic acid magnetic particles (Thermo Fisher) in the presence of EDC (250 mM), imidazole chloride (1 mM) and NaCl (200 mM). The mixture was incubated overnight at room temperature. The aptamer particles were then passivated by conjugation with amino-PEG12 (Pierce Biotechnology, 2 mM) by incubating with EDC (250 mM) and sulfo-NHS (100 mM) in PBS buffer for 16 h at room temperature. After passivation, the particles were washed twice for 30 min with 1 ml of TT buffer (50 mM Tris, 0.1% Tween 20, pH 8.0), and then resuspended in TET buffer (10 mM Tris, pH 8.0, 0.1 mM EDTA, 0.1% Tween 20) at a concentration of $10^7$/μl and stored at 4 °C.

Next, monoclonal aptamer particles were generated through emulsion PCR. The oil phase was composed of Span 80, Tween 80, and Triton X-100 (Sigma-Aldrich, 4.5%, 0.40%, and 0.05%, respectively) in mineral oil. The aqueous phase consisted of Taq PCR Master Mix (Promega, 1×), MgCl₂, reverse primer (RP, 3 μM), GoTaq Hot Start Polymerase (Promega, 0.5 U/μl), template DNA (2 pM)FP-coated particles (3×10⁸) in a total volume of 1 ml. Water-in-oil emulsions were prepared by adding 1 ml of the aqueous phase to 7 ml of oil phase in a DT-20 tube (IKA) locked into the Ultra-Turrax Device (IKA). This addition was performed drop-wise over 30 s while the mixture was being stirred at 650 RPM in the Ultra-Turrax. Then 100 μl aliquots of the emulsion were pipetted into ~80 wells of a 96-well PCR plate. We performed PCR under the following cycling conditions: 95 °C for 3 min, followed by 50 cycles of 95 °C for 15 s, 60 °C for 30 s and 72 °C for 75 s.

After PCR, the emulsion was broken by mixing 50 μl of 2-butanol with 100 μl of the PCR reagent mixture in each well. The broken emulsions were combined into a 50 ml tube, vortexed, and centrifuged at $2500 \times g$ for 5 min. After carefully removing the oil phase, the particle pellet was resuspended with 1 ml of single-strand generation (SSG) buffer (100 mM NaOH, 100 mM NaCl, 1% Triton X-100, 10 mM Tris-HCl, pH 7.5, and 1 mM EDTA), transferred to a new 1.5 ml tube and incubated at 50 °C for 2 min. The particles were then pelleted via a magnetic separator and the supernatant was removed. The particles were washed two more times with SSG buffer using magnetic separation, then resuspended in 300 μl TE.

To characterize the monoclonality and the density of the aptamer particles, AlexaFluor 488-labeled RP was annealed with 10⁶ aptamer particles in TE buffer at 55 °C for 10 min and snap-cooled on ice for 2 min. The particles were then washed twice with 100 μl TE buffer and analyzed by flow cytometry.

**PD screening**. When incubated with the target protein, a fraction of the aptamers on each particle bind the target. The binding fraction (Fr) is dictated by the aptamer $K_D$, and the concentration of the target ([T]), with $Fr = [T]/([T] + K_D)$. Fr can be directly quantitated by fluorescence-activated cell sorting (FACS) by the intensity (F) at a given target concentration with $Fr = F/F_{max}$. Thus, to isolate aptamers with a desired $K_D$ threshold, we incubated with a target concentration of $K_D/2$ and set the FACS collection gate to $\geq F_{max}/3$ (Fig. 2a). In this way, we collected pools with sharply defined affinity thresholds and minimal false positives and negatives[29]. It is important to note that the absolute $K_D$ value of the threshold is contingent upon consistent $F_{max}$, floruenccenic channel gains and target concentration, which can vary from experiment to experiment. Since it is essential to compare the absolute $K_D$ threshold between experimental sets (PD and MLPD), after the pools were collected and sequenced, we calibrated the $K_D$ thresholds by retrospectively analyzing a subset of aptamers within each bin (detailed below in the aptamer characterization section).

During each round of screening, we incubated ~$10^8$ aptamer particles with NGAL (R&D systems, C-terminal His-tag) at different concentrations (64, and 256 nM for Rounds 1; 16, 64 and 256 nM for rounds 2; 4, 16, 64, and 256 nM for the ML generated library) in 1 ml of PBSMCT buffer (DPBS with 2.5 mM MgCl₂, 0.9 mM CaCl₂, 0.01% Tween 20)[52]. His-Tag peptide (GenScript, 10 μM) was included to avoid generating aptamer against the His-Tag attached to NGAL. After incubating for 1 h at room temperature, the unbound NGAL was washed away with PBSMCT via magnetic separation. To fluorescently label the bound NGAL, iFluor 647 His-Tag antibody (Genscript, 6 nM) was added to the mixture and incubated for 30 min. The particles were magnetically washed with PBSMCT to remove the excess antibody and resuspended in PBSMCT buffer.

The samples were then analyzed via FACS (SONY SH800) with the sorting threshold set to 1/3 of the maximum fluorescent intensity ($F_{max}$), which was determined by incubating a positive control aptamer[29] at a saturating concentration of NGAL (1 μM) and fluorescently labeling as described above. Particles with intensity greater than $F_{max}/3$ were collected as a positive population and particles with intensity lower than $F_{max}/3$ were collected as a negative population. The isolated aptamers were amplified by GoTaq DNA polymerase to generate an enriched pool for a subsequent round of aptamer particle synthesis or to append sequencing adapters to perform next-generation sequencing (NGS) as described below.

## Sequencing and data processing

*Sequencing of PD screening pools*. To perform NGS, adapter sequences were first appended to the sequences contained in each pool. Overhang adapter sequences for the forward and reverse primers were synthesized at IDT. The Illumina P5 adapter was added to the 5′ end of FP (P5FP) and a hexamer index and P7 adapter was added to the 5′ end of the RP (P7RP). Both positive and negative populations from each NGAL concentration were indexed by performing PCR with respective mixtures of P5FP (1 μM), P7RP (1 μM), dsDNA template (0.1 nM) and 1× KOD Hot Start master mix. PCR products of the correct size were purified via agarose gel extraction using QIAquick Gel Extraction Kit. Each sample was quantified via Bioanalyzer and mixed at equimolar ratio for sequencing on the NextSeq platform. The number of reads for each sequencing pool can be found in Supplementary Table ST9.

*Data preprocessing*. To prepare the sequencing data for model training, the DNA sequences in the raw fastq files were filtered to remove low-quality reads and clustered to group similar sequences. Conceptually, for the clustering, we created a graph where sequences (nodes) were linked if they were within a Levenshtein distance of five. Cluster ids were assigned to each connected component in the graph. An efficient implementation of this approach is described below. In practice it enabled the clustering of up to 1 billion reads. Once clusters were identified, we trained the models in one of three ways: (1) with all the sequences, but keeping all sequences in a cluster within one partition; (2) only the "cluster representative" (sequence within the cluster with the highest count) kept; or (3) only the "cluster representative," but the value for this representative was the sum of all the sequence counts in the cluster. In practice, these models performed similarly, though the models trained faster when only using cluster representatives. We, thus, used cluster representatives for selecting sequences in MLPD. Lastly, before use in any ML models, sequencing counts were normalized to the total number of sequences from the experiment. The sequences were divided into five equal folds and the first fold (20% of the data) was selected to be the test set, and the models were trained on the remaining 80% of the data.

*Clustering approach*. The conceptual graph algorithm for identifying clusters is the following:

1. Add all unique sequences as nodes to a graph.
2. For each pair of nodes, calculate Levenshtein distance and add an edge to the graph if the distance is <= 5.
3. Assign a unique cluster ID to each connected component of the resultant graph.

However, because the input data may be quite diverse (>1 billion sequences), the naive algorithm presented above is intractable. We implement an approximation of the above algorithm in the following way:

First, we split the input data into two distinct subsets. One subset was used for an all-pairs comparison to define initial clusters, and the other was projected into the clusters defined by the all-pairs subset. To maximize the accuracy of all-pairs cluster detection, its input sequences were multi-read sequences; i.e. those for which the sum of all read counts in all conditions is greater than one. If this subset of sequences is sufficiently small, the subset is then padded with singleton sequences until either the entire dataset is exhausted or its size is as large as is computationally tractable. In practice, we set the size of the all-pairs subset to 300,000,000.

Second, we converted the two subsets of sequences to vectors representing the counts of all 6-mers in each sequence. These feature vectors were used to identify candidate neighbors, both for the all-pairs subset (comparing it to itself) and the projection subset (projecting each of those against the all-pairs data) using an inverted index algorithm and cosine distance[53,54]. Finally, we filtered the candidate neighbors to the set of true neighbors by performing explicit calculation of Levenshtein distance. We used the true neighbors in the all-pairs subset to generate a set of candidate clusters, and then added the projected sequences into those clusters.

**ML models, features, and output layers**. We constructed two representations of the aptamer sequence: a one-hot encoding and kmer-counts encoding. In the one-hot encoding, each nucleotide choice is represented as a binary value among the 4 possible bases; at a given position in the sequence, the observed aptamer nucleotide is set to 1 and all other nucleotides are set to 0. The resulting vector contains 160 binary values. To provide direct, local contextual information, we also encode counts for all $k$-mers up to length 4. The 340 additional values lead to a combined input vector length of length 500.

A challenge in the model design was determining the best representation for the output layers. The goal of the network was to predict the affinity of an aptamer sequence to a target, but the training data was sequence counts from particle display or these same counts binned into binary values, which we assumed to be correlated with, but not an exact measure of, affinity. We implemented two approaches. The fully connected approach directly connected the training count or bin value to the hidden NN layers. The latent affinity approach, used only for the Count model, added an affinity prediction layer between the fully connected NN layers and the sequence count output, with each target affinity value connected to all the sequence count outputs for that target. The motivation was that the hidden layers would learn to predict an affinity for each target, and this affinity value, multiplied by a single trained float parameter per output, would predict all the sequence counts. The hope was that squeezing the predictions through this latent affinity representation would create a model better able to predict affinity across multiple conditions, though in practice the fully connected and latent affinity approaches showed similar performance. Details of the selected models are described below and diagrammed in Supplementary Fig. S6.

*Counts ML model*. The ML models predicted affinity by predicting the results of the PD experiments, processed in one of three ways. The first model trained was the Counts model, which directly predicts the normalized sequence count in each positive and negative pool. For fully connected models, we then calculated affinity to a target as the sum of one or more sequence count outputs, here, the sum of the round 264 nM and 128 nM positive pools. For the latent affinity models, the affinity values are the latent affinity prediction layer. While multiple NN architectures were tested, in combinations of 0–3 convolutional layers followed by 0–3 fully connected layers, a naive fully-connected neural network with or without convolutions performed well in practice. We used Vizier[41] testing 100 potential models via a random search, to identify optimal hyper-parameters. Specifically, we explored the following hyper-parameters: number of hidden layers, number of convolutional layers, choice of activation function nonlinearity, learning rate, momentum, drop-out probability, balancing positives, and size of the hidden layers. The models were ranked by the AUC over the top one percent of non-zero output values in the test set. The selected Count model uses the latent affinity output architecture, squared error loss, mini-batch size of 64, a learning rate of 0.00138, and momentum of 0.903 (Supplementary Fig. S6). We upsampled positive sequences (those with a summed count of at least 1000 across all rounds) so they made at least 10% of each training batch. In addition to the basic sum of counts, the Counts model variant also encouraged the model to simultaneously learn inherent properties of aptamers that were independent of the target and could be determined directly from the sequence. Secondary structure was a logical choice, as its impact on aptamer affinity is well-established[34]. The intuition was that by simultaneously learning structure the model could encode biological properties that would inform its protein target predictions. Rather than a complex secondary structure representation, we used the partition function, a single floating point value that aggregates the energy associated with all possible structures weighted by their Boltzman probability[55].

*Binned ML model*. The Binned model variant simplified the data representation from the underlying particle display experiment. By design, the PD experiment should yield two pools of sequences (positive and negative) at each target concentration. This implies that each sequence could be represented with an array, $b$, of binary labels, where the value at each position, $b_i$, in the array indicates if the

sequence was in the "positive" pool at concentration $c_i$. In practice, a small number of sequences were not clearly distinguishable as positive or negative, to account for this we created a ternary label. The Binned model simply replaced the sequencing counts with the sequence's label in each of these stringency bins. Additionally, both the Binned and SuperBin models used the AdamOptimizer with a learning rate 0.001. The selected Binned model uses the fully connected output architecture, squared error loss, a learning rate of 0.00388, and momentum of 0.737 (Supplementary Fig. S6).

*SuperBin ML model.* The SuperBin representation attempted to summarize all PD experiments with a single number. Naively, we can consider a sequence present in the positive pool at all tested affinities to have strictly higher affinity than one present in only medium and low stringency, positive pools (which are similarly higher than one present in zero or only the low stringency positive pool). In practice, we observed that there are very few positive sequences, substantial ambiguous information (from low sequencing counts), and conflicting labels (e.g. sequences that appear in a positive bin at affinity $a_j$, that were in a negative bin for some $b_i$ where $i < j$). We therefore created a final target representation, in which we proposed seven levels of affinity which tries to take into account this ranking as well as potential "borderline" assignments. Sequences containing conflicting or ambiguous information were simply eliminated. The selected SuperBin model uses the fully connected output architecture, squared error loss, a learning rate of 0.00203, and momentum of 0.498 (Supplementary Fig. S6).

**ML derived aptamer sequences.** Input seeds were derived both from existing particle display aptamers (experimental seeds) as well as random sequences predicted to be high-affinity by an ML model (termed "ML seeds" to differentiate them from truly random seeds used without an ml pre-screening step). For the ML seed sequences, up to 1 billion random aptamers were generated and scored via the model. The top 5000 (0.0005%) sequences from each model were selected and used as input for walking.

We applied a straight-forward iterative mutation strategy to locally improve each seed sequence. In short, we first created a "parent set" that contained only the initial seed sequence. We then performed model inference on 10,000 random mutations, each up to 4 substitutions away from sequence(s) in the parent set. The top five sequences were selected for experimental validation and the top 200 mutants become the parent set for the next round of walking. This process was repeated for five rounds.

The distribution of mutational distances between the generated sequences and the input seeds was similar across the different models (Supplementary Fig. S5). The generated aptamers, as well as their seeds, were synthesized and screened on particle display at four different NGAL concentrations (the number of synthesized sequences for each category is shown in Supplementary Table ST3). Positive and negative pools were collected, sequenced, and validated as described above (Fig. 2C, Supplementary Table ST1). Figure 3 shows the model-guided exploration performance.

*Computationally predicting core sequences.* For each full-length sequence, we examined all subsequences of at least 15 nt to estimate their putative efficacy as a reduced length aptamer. Within the constrained 40 nt interval evaluated by the SuperBin model, we placed each subsequence at every available start index. At this index, we flanked the subsequence by prefix and suffix sequences consisting entirely of "A," "C," "G," and "T"; in total, this process generated $4 * (40−l+1)$ aptamer variants, where $l$ corresponds to the length of the subsequence. All variants were scored by the model, yielding a distribution for each subsequence (Fig. 4). Candidate sequences for validation were identified for testing by calculating the median and variance for subsequences of each length (15, 19, 23, 27, 31, 35, and 39 nucleotides), and selecting candidates that maximized the median model score or minimized variance. We also selected sequences with high low model score and high variance to verify the effect of these parameters.

**Characterization of individual full-length and truncated core aptamers.** In order to experimentally evaluate the binding performance of individual aptamer sequences, we synthesized candidates to determine their affinity via a bead-based fluorescence binding assay[29].

*Preparation of aptamer particles for affinity measurement.* For affinity measurement of all full-length aptamer sequences, each individual sequence was synthesized and purified by Integrated DNA Technologies (IDT) without modification. To prepare aptamer particles for the fluorescence binding assay, aptamers were coated on to forward primer-conjugated particles via PCR and converted to single-stranded aptamer as previously described[29].

For affinity measurement of aptamer core sequences which do not have primer regions, all individual sequences were synthesized by Integrated DNA Technologies (IDT) with a biotin conjugated to the 5′ end. To prepare aptamer particles for the fluorescence binding assay, 1 μM of each biotinylated aptamer was first incubated with $10^7$ MyOne Streptavidin C1 (Thermofisher) particles in 100 μl PBSMCT buffer (DPBS with 2.5 mM $MgCl_2$, 0.9 mM $CaCl_2$, 0.01% Tween 20) for 30 min at

room temperature. The aptamer particles were then washed in PBSMCT via a magnetic separator to remove any unbound aptamer and then resuspended in 100 μl PBSMCT.

*Full $K_D$ measurement from aptamer particles.* To determine the accurate $K_D$, different concentrations of NGAL protein (1–1024 nM at twofold increment or 1 nM to 1 μM at 3.16-fold increment or 10 nM to 10 μM at 3.16-fold increment) were first incubated with a fixed amount of the aptamer-coated particles ($10^4$ particles/l) in 100 μl PBSMCT for 1 h at room temperature. The unbound NGAL was magnetically washed away with PBSMCT. To fluorescently label the bound NGAL, 6 nM iFluor 647 His-Tag antibody (Genscript) was introduced to the mixture and incubated for 30 min. The particles were magnetically washed with PBSMCT and the median fluorescence intensities were quantified via FACS (BD Accuri C6) at each concentration of NGAL. $K_D$ was derived using GraphPad Prism by applying the single-site binding model.

*Approximate $K_D$ measurement.* To determine approximate affinity, four concentrations of NGAL protein (4, 16, 64, and 256 nM) were tested using the same approach as described above. The median fluorescence intensities of the bound complex were quantified via FACS (BD Accuri C6) at each concentration of NGAL and compared to the maximum fluorescence intensity ($F_{max}$). To assign the estimated $K_D$ value to a given aptamer sequence, the following metric was applied. For each concentration tested, the estimated $K_D$ was assigned if the median intensity was $\leq F_{max}/5$ ($K_D > 4[T]$), $> F_{max}/5$ but $\leq F_{max}/3$ ($2[T] < K_D < 4[T]$), or $> F_{max}/3$ ($K_D < 2[T]$).

*Secondary structure analysis.* DNA structures analysis was performed using the ViennaRNA package (v2.4.13)[56]. Free energies for individual sequences were calculated at 37 °C using DNA parameters (Matthews model, 2004). The structure with the lowest free energy for the full-length and 23 nt truncation of G12, G13 were plotted in Fig. 4. To assess if the TGGATAG was observed in a loop, we visually inspected the top 10 lowest free energy predictions. Supplementary Table ST3 identifies aptamers in which any of these predictions showed the motif completely contained within a hairpin.

**Reporting summary.** Further information on research design is available in the Nature Research Reporting Summary linked to this article.

## Data availability

Analyzed datasets generated during the current study are available in the Supplementary Information and via the github repository at https://github.com/google-research/google-research/tree/master/aptamers_mlpd. Raw sequencing reads are available in the NCBI SRA repository, PRJNA672779. Source data are provided with this paper.

## Code availability

Code for analysis of the provided datasets is provided as runnable IPython notebooks at https://github.com/google-research/google-research/tree/master/aptamers_mlpd.

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

## Acknowledgements

We would like to thank Shuo Yang for generating the Fig. 1 schematic.

## Author contributions

M.B., M.D., Q.G., J.W., and B.S.F. conceived the study. Q.Y., A.S., J.J., H.K., W.C. Q.G., J.W., and B.S.F. performed particle display, aptamer characterization and analyzed the results. A.P. sequenced aptamers. A.B., Z.A., G.E.D, G.D, S.H., C.M., and M.D. wrote and trained ML models and analyzed the results. A.B. and M.D designed sequences used in MLPD. A.B., M.D., C.M., Q.Y., and B.S.F. wrote the manuscript. All authors read and provided comments on the manuscript.

## Competing interests

Q.Y., W.C., Q.G., J.J., H.K., A.S., J.W., and B.S.F. are employees and/or shareholders of Aptitude Medical Systems Inc. A.B., S.H., C.M., G.D., Z.A., A.P., G.E.D., M.B., and M.D. are employees of Google, a technology company that sells machine learning services as part of its business.
