## [Peer Review File · Nature Communications]

Reviewers' Comments:

Reviewer #1:

Remarks to the Author:

1. The first sentence of the abstract should clearly define/describe an aptamer. Overall, recall that Nature Communication has a broad readership.

2. The abstract should be explicit that these are DNA aptamers. The manuscript text also needs to be explicit that these are DNA aptamers. The authors' description of their aptamers' physical/chemical properties is only correct if they are specific to DNA aptamers and not RNA aptamers, which are more thermolabile and have lower stability. The authors need to update their manuscript text throughout.

3. The first section on "MLPD: Machine Learning Guided Particle Display" is not informative enough and seemingly jumps from model training to using the model for sequence design in the space of 2 sentences. If this is an overview of the entire experimental/computational process, it should more closely mirror the order shown in Figure 1 without going into methodological detail in this section. But the purpose of each step should be clearly explained.

4. The authors need to briefly explain the methodology to carry out Particle Display in the main text, focusing on the key steps [e.g. the use of a fluorescent dye, the flow-assisted cell sorting binning process] and the data outcomes. The authors need to more clearly explain what it means by "increasing stringency 4-fold" [by 4-fold]. Also, it is not recommended to overly use the abbreviation PD in figure legends.

5. Why did the authors select the threshold as $F_{max}/3$? How did F_{max} change when varying target (NGAL) concentration? How many aptamer variants passed the threshold in each experiment? Is this considered a ML problem with Class Imbalance; how did the authors handle that?

6. What NGS was performed (MiSeq, HiSeq, etc)? How many mapped reads were obtained in each round of experiments? In Figure 2, do the Venn diagrams show the number of unique aptamer sequences observed in each bin/category? How many unique aptamer sequences were observed? How many unique aptamer sequences within 1 edit distance of mismatch? There is a great deal of information in the Supplementary Tables, but these details need to appear in the main text with corresponding citations to Supp Tables. Overall, the figures are very clear and informative; however, the manuscript text needs to more explicitly mirror the process shown in the figures as well as provide more methodological and outcome details. For example, it is important in the main text to highlight key points & conclusions using numbers shown in the figures.

7. When defining the criteria for labeling an aptamer as part of the positive pool, it's important for the authors to provide a clear example with actual numbers. What was 20% of the expected value of a single particle display bead? What was considered a typical prevalence in the positive pool vs. the negative pool, in terms of % of mapped reads within each pool?

8. In the section on "Trained ML models can predict high affinity aptamers in a held-out dataset", the manuscript main text needs to more clearly explain the different overall approaches used to develop and train CNN models. Then, after the best encoding and best CNN layout was selected and carrying out hyperparameter optimization, the manuscript text needs drill down and explain the numerical details on the best trained models. For example, the "Counts" model is a regressor whereas the "Binned" and "SuperBin" models are classifiers; this distinction should be clearer in the main text.

One-hot encoding and kmer encoding were both used; which one worked better? [The main text is also not clear that they were used separately and not at the same time].

Typical 80/20 splits were used for CNN training. Was K-fold cross-validation used, which is routine

nowadays?

What did the trained CNNs look like? What filters were used? How many nodes per layer? How many layers?

9. In the section "Trained ML models can predict novel aptamers with high affinity", the sequence design procedure & results are convoluted. If the purpose was to design a set of aptamer sequences with the highest model scores, then the optimization algorithm should be run until the model score is maximized, per each initial/seed sequence. This is a standard application of "Monte Carlo optimization" [also known as Simulated Annealing] in the optimization field. If the purpose was to design a larger set of aptamer sequences with varying model scores that more broadly span the sequence space, then sampling sequence during the "walking" of sequence space could make sense. Right now, the manuscript text describes an ad hoc algorithm for generating sequences, starting from different types of initial sequences/seeds, but it's not clear if these sequences are actually predicted to have higher affinities according to the CNN model.

For example, as described, mutating a random sequence by 2 nucleotides is not ever expected to generate a valid aptamer. Therefore, a conclusion that "the ML directed walks substantially outperform random walks and the original PD" doesn't have value; in the optimization world, you would simply iterate an efficient optimization algorithm until an optimal solution is obtained.

The important & valuable key question is whether the CNN model is able to predict aptamer affinity accurately enough to significantly narrow down the experimental sequence space that needs to be tested. Figure 3 does not answer that question, but the authors could likely answer it with their existing dataset. The authors need to compare the CNN model predictions to aptamer characterization measurements to answer this question. It could be a regression or classification test, but it must clearly show the number of aptamers (exceeding each measured affinity threshold) that were predicted by a CNN model to bind to their target. As an important comparison, the authors can also utilize the CNN model to predict when an aptamer sequence is NOT expected to have appreciable affinity to the NGAL target. Here, the authors can utilize their "Random Seed" as the negative control for this comparison.

10. The conclusion "This demonstrates that MLPD can yield better aptamers than the examples it was trained with." looks to be supported by the authors' data, but the described comparisons between an "experimental seed" or a "ML seed" or a "randomly screened ML seed" don't make sense. When considering the usage of the CNN predictions to extrapolate beyond the training set, the key question is how extrapolative accuracy depended on the training set sequences. For example, the CNN model's predictions might be equally accurate on the entire aptamer sequence space or perhaps only a subset of the aptamer sequence space that is "closest" (by some metric) to the training sequence set. It would be worthwhile and interesting to quantify (e.g. using minimum edit or Hamming distance) the distance between the newly designed aptamer sequences and training set sequences and compare this distance to the CNN model accuracy. This is distinct from what is shown in Figure S5, which shows the edit distance between the designed sequence and the seed sequence.

11. The authors don't provide any statistical analysis to support the importance of the GTGGATAG motif or indicate the location of this motif within the overall aptamer. There are always huge "edge effects" when randomly varying a sequence within a larger sequence region that could be biasing the motif importances. Unless the authors can provide more support for the importance of this motif, it should be removed from the text.

12. The authors must carry out a structural similarity analysis to determine if all the high-affinity DNA aptamers fold into same/similar DNA structures. DNA structure plays a key role in creating the binding pockets for target binding. If the high affinity aptamers folded into similar structures, it would provide more evidence of their functionality. This analysis should be extended to the truncated DNA aptamers

to determine if the truncated portion corresponds to a particular subset of the structure. As it stands, besides the K_d measurements, there is no physical/chemical analysis of the DNA aptamer sequences that explains WHY they bind to the target.

13. When discussing the characterization of the truncated aptamers, the author should refer to the "seed" sequences as full-length sequences. The authors are using the word "seed" many times for different purposes and changing the definition.

14. In the introduction, the authors need to explain what the NGAL target is and why it's generally important/relevant as a biomarker.

Reviewer #2:

Remarks to the Author:

this is a useful manuscript which should be published

REVIEWER RESPONSES

We thank the reviewers for their comments and have responded to each inline below (original comment in italics, our response in bold). We have incorporated many of the reviewers' suggestions which we feel has substantially improved the text. We have included a version of the manuscript with all changes tracked, some of the new text has also been included in the responses for ease of review.

In addition, we have added one author, Hui Kang, who performed the secondary structure analysis for the revision. We have added corrections to the figures and tables not related to reviewer responses:

- Figure 3, we changed the title of Panel A to be consistent with the caption.
- Table ST4, the original table repeated a line, we have corrected the numbers for the Binned walks.
- Figure S3: Corrected pie-chart, resulting in slight affinity changes (+/- 1 level) for 2 of the 385 aptamers.

Reviewer #1:

1. The first sentence of the abstract should clearly define/describe an aptamer. Overall, recall that Nature Communication has a broad readership.

We have more clearly defined aptamer in the first sentence as well as in the introduction.

Abstract:

“Aptamers are single-stranded nucleic acid ligands comprised of DNA, RNA or chemically-modified variants that bind with high affinity and specificity to target antigens.”

Introduction:

“Aptamers are single-stranded nucleic acid ligands that can be developed to bind a wide range of targets with high affinity and specificity. Comprised of DNA, RNA or chemically-modified nucleic acids, ...”

2. The abstract should be explicit that these are DNA aptamers. The manuscript text also needs to be explicit that these are DNA aptamers. The authors' description of their aptamers' physical/chemical properties is only correct if they are specific to DNA aptamers and not RNA aptamers, which are more thermolabile and have lower stability. The authors need to update their manuscript text throughout.

We have updated the abstract and multiple places within the manuscript text to reflect that DNA aptamers are used in this proof-of-concept study.

Also the reviewer is right that natural RNA aptamers are usually not sufficiently stable for therapeutic or diagnostic applications, so we updated the introduction to clarify accordingly. We do note that RNA aptamers can be chemically modified to attain sufficient stability for such applications and this is routinely done in therapeutic aptamer development. Finally, we have also clarified that the Machine-Learning Particle Display approach described herein is agnostic with regard to the composition and hence can be applied to aptamers comprised of DNA, RNA, and modified DNA/RNA.

3. The first section on “MLPD: Machine Learning Guided Particle Display” is not informative enough and seemingly jumps from model training to using the model for sequence design in the space of 2 sentences. If this is an overview of the entire experimental/computational process, it should more closely mirror the order shown in Figure 1 without going into methodological detail in this section. But the purpose of each step should be clearly explained.

We thank the reviewer for the suggestion. We’ve expanded the description in the beginning of the Results. Additionally, we now explicitly reference each subsection of the figure and provide the purpose and/or motivation of each step.

4. The authors need to briefly explain the methodology to carry out Particle Display in the main text, focusing on the key steps [e.g. the use of a fluorescent dye, the flow-assisted cell sorting binning process] and the data outcomes. The authors need to more clearly explain what it means by “increasing stringency 4-fold” [by 4-fold]. Also, it is not recommended to overly use the abbreviation PD in figure legends.

We agree with the suggestion. We included the following brief description of the methodology in the main text and modified figure legends to reduce the abbreviations usage.

“To perform a round of PD, the pool of aptamer particles was first incubated with fluorescently-labeled target at a given concentration. Next, the pool was screened via fluorescence-activated cell sorting (FACS) to partition the particles based on a gating fluorescence value as the threshold (see Methods for details). The affinity thresholds were separated by 4-fold each by lowering the target concentration 4-fold while keeping the same sorting gate value at $F_{max}/3$ (Supp. Table ST1).”

5. Why did the authors select the threshold as $F_{max}/3$? How did F_{max} change when varying target (NGAL) concentration? How many aptamer variants passed the threshold in each experiment? Is this considered a ML problem with Class Imbalance; how did the authors handle that?

We selected a threshold of $F_{max}/3$ to maximize the accuracy of our affinity-based gating. As a threshold, it yields the maximum difference in fluorescence signal between an aptamer particle with a desired affinity and any aptamer particle that has worse K_D . For a detailed explanation, we have cited our previous work that provides the comprehensive theoretical analysis (2014 Particle Display: A Quantitative Screening Method for Generating High-Affinity Aptamers).

F_{max} didn't change when varying target concentrations. F_{max} represents the saturation state that all the aptamers on the particle are bound to a target. Thus, F_{max} only depends on 1) the amount of the aptamer per particle and 2) the fluorescent intensity of the detection reagent. Both factors are independent of the target concentration.

We have added the number of unique clustered sequences in each selection to Supplemental Table ST1. There is a class imbalance with many more negative sequences than positive ones, especially at the high stringencies. For the Counts model, we investigated balancing the training by ensuring at least a minimal number of positive sequences within each training batch. In specific, we upsampled sequences with sum total count values > 1000 until they made up 10% of the batch. This option seemed to slightly improve AUC scores. Though the difference was small, this option was left on during training of the Counts model, and we've added details in the Method section. For the Binned and SuperBin models, we did not explicitly balance the positives per training batch, though it would be possible.

This is one of the reasons we discuss the potential for iterative approaches in the future. In this work we show that we can generate many more positive sequences, and we believe iteratively making, and experimentally measuring, predictions has the potential to improve models further.

6. What NGS was performed (MiSeq, HiSeq, etc)? How many mapped reads were obtained in each round of experiments? In Figure 2, do the Venn diagrams show the number of unique aptamer sequences observed in each bin/category? How many unique aptamer sequences were observed? How many unique aptamer sequences within 1 edit distance of mismatch? There is a great deal of information in the Supplementary Tables, but these details need to appear in the main text with corresponding citations to Supp Tables. Overall, the figures are very clear and informative; however, the manuscript text needs to more explicitly mirror the process shown in the figures as well as provide more methodological and outcome details. For example, it is important in the main text to highlight key points & conclusions using numbers shown in the figures.

Answers to questions in line:

What NGS was performed (MiSeq, HiSeq, etc)?

- The NGS performed was NextSeq. This is in the methods, but we have added it to the main text:

“By varying the target concentration to control the stringency, PD was used to partition this library into positive and negative aptamer pools at multiple affinity thresholds, each of which was characterized via Next Generation Sequencing (NGS) on the Illumina NextSeq (Methods: Sequencing and data processing).”

How many mapped reads were obtained in each round of experiments?

- **Because the library is unknown to start, we do not have a notion of mapped reads. However, ST7 reports the total number of reads and reads passing for each round, stringency, and positive/negative pool. Additionally, ST1 now includes the total number of unique clusters for each round, stringency, and pool.**

In Figure 2, do the Venn diagrams show the number of unique aptamer sequences observed in each bin/category?

- **All numbers are the number of unique aptamers after clustering (not just unique sequences, but unique clusters).**

How many unique aptamer sequences were observed? How many unique aptamer sequences within 1 edit distance of mismatch?

- **We think the closest estimate to this is the number of unique clusters (so sequencing errors have been collapsed). We have added the number of unique aptamer sequences to ST1. For the original PD experiment, there were 910,441 unique clusters across all the rounds of sequencing. We detected 8,617,439 unique sequences that were an edit distance of 1 from the cluster representative and 2,263,634 that were at edit distance 2.**

7. When defining the criteria for labeling an aptamer as part of the positive pool, it's important for the authors to provide a clear example with actual numbers. What was 20% of the expected value of a single particle display bead? What was considered a typical prevalence in the positive pool vs. the negative pool, in terms of % of mapped reads within each pool?

What was 20% of the expected value of a single particle display bead?

Below is a table showing the # of reads required to achieve 20% of the expected bead coverage within each experiment.

Experiment	Stringency	# of reads required to achieve 20% expected bead coverage (positive)	# of reads required to achieve 20% expected bead coverage (negative)
Original PD	< 2 uM	1373	1170
Original PD	< 512 nM	1469	440

Original PD	< 128 nM	1311	575
MLPD	< 512 nM	26	26
MLPD	< 128 nM	59	55
MLPD	< 32 nM	204	192
MLPD	< 8 nM	374	387

We have modified the corresponding sentence in the text that references the 20% cutoff: “First, the aptamer had to be reliably detected in the positive pool, defined as having a sequencing count of at least 20% of the expected bead coverage (corresponding to 1373, 1469, 1311 sequencing counts for the 2 μ M, 512 nM, and 128 nM stringencies, Supp. Table ST2).”

What was considered a typical prevalence in the positive pool vs. the negative pool, in terms of % of mapped reads within each pool?

For each circle in the venn diagram we computed the mean % of reads within each pool. Sequences from within a stringency level are labeled as “Passing Stringency” with the sequences outside the stringency level (as indicated in the Venn Diagram) labeled as “Other” sequences. For sequences in positive pools the counts were (on average) >30X the counts in the negative pool for all conditions. While sequences in the “other” pool typically had 5-10X more sequence counts in the negative pool.

Expt.	stringency	Passing Stringency			All Other Sequences		
		positive pool mean fraction	negative pool mean fraction	positive mean frac / negative mean frac	positive pool mean fraction	negative pool mean fraction	positive mean frac / negative mean frac
Original PD	low	0.0005	0.000015	32.500157	0.000002	0.000019	0.105069
Original PD	medium	0.0008	0.000024	33.513512	0.000003	0.000027	0.096864
Original PD	high	0.001804	0.000043	42.10854	0.000006	0.00003	0.190722
MLPD	low	0.000097	0.000002	45.58161	1.13E-06	0.00001	0.114675
MLPD	medium	0.000295	0.000006	45.948491	6.63E-07	0.000011	0.061902
MLPD	high	0.002244	0.000077	29.016303	5.25E-07	0.00001	0.054673
MLPD	very	0.008374	0.000209	40.051441	6.56E-07	0.000009	0.07051

We modified the following sentence in the text:

“All positive pool counts, on average, exceeded negative pool counts by >29X for sequences passing a stringency threshold.”

8. In the section on “Trained ML models can predict high affinity aptamers in a held-out dataset”, the manuscript main text needs to more clearly explain the different overall approaches used to develop and train CNN models. Then, after the best encoding and best CNN layout was selected and carrying out hyperparameter optimization, the manuscript text needs drill down and explain the numerical details on the best trained models. For example, the “Counts” model is a regressor whereas the “Binned” and “SuperBin” models are classifiers; this distinction should be clearer in the main text.

One-hot encoding and kmer encoding were both used; which one worked better? [The main text is also not clear that they were used separately and not at the same time].

Typical 80/20 splits were used for CNN training. Was K-fold cross-validation used, which is routine nowadays?

What did the trained CNNs look like? What filters were used? How many nodes per layer? How many layers?

We agree with the reviewer’s comment and have added a graphical depiction of the model structure used for all models (Supplemental Figure S6). In particular we have noted the variable parameters and architecture components that were optimized with Vizier (that are described explicitly in the methods), so that the methods descriptions are easier to understand.

Additionally, we have added text in response to the reviewers comments:

- **We have clarified that all models are regressors, adding this sentence to the results: “In all three prediction tasks, the model is trained with least-squares regression.”**
- **Specifically, both the one-hot encoding and the kmer-encoding were used in all. We have added the following sentence to the results: “For the input, our models used the concatenation of a simple one-hot and kmer count-based representation of the input sequence (Methods, *ML Model Design*).”**
- **We used a 80/20 split for CNN training, but did not use K-fold cross-validation. Our hope was that the experimentally-validated forward predictions would be the real proof of the model’s utility. However, we have added this point to the discussion as a limitation of our strategy and an area for improvement. “Notably, a limitation of our approach was the use of a single test set (holding out 20% of the data) for hyperparameter tuning. Given the limited positive examples, cross-validation could improve robustness of the tuned models.”**

9. In the section “Trained ML models can predict novel aptamers with high affinity”, the sequence design procedure & results are convoluted. If the purpose was to design a set of aptamer sequences with the highest model scores, then the optimization algorithm should be run until the model score is maximized, per each initial/seed sequence. This is a standard

application of “Monte Carlo optimization” [also known as Simulated Annealing] in the optimization field. If the purpose was to design a larger set of aptamer sequences with varying model scores that more broadly span the sequence space, then sampling sequence during the “walking” of sequence space could make sense. Right now, the manuscript text describes an ad hoc algorithm for generating sequences, starting from different types of initial sequences/seeds, but it’s not clear if these sequences are actually predicted to have higher affinities according to the CNN model.

For example, as described, mutating a random sequence by 2 nucleotides is not ever expected to generate a valid aptamer. Therefore, a conclusion that “the ML directed walks substantially outperform random walks and the original PD” doesn’t have value; in the optimization world, you would simply iterate an efficient optimization algorithm until an optimal solution is obtained.

The important & valuable key question is whether the CNN model is able to predict aptamer affinity accurately enough to significantly narrow down the experimental sequence space that needs to be tested. Figure 3 does not answer that question, but the authors could likely answer it with their existing dataset. The authors need to compare the CNN model predictions to aptamer characterization measurements to answer this question. It could be a regression or classification test, but it must clearly show the number of aptamers (exceeding each measured affinity threshold) that were predicted by a CNN model to bind to their target. As an important comparison, the authors can also utilize the CNN model to predict when an aptamer sequence is NOT expected to have appreciable affinity to the NGAL target. Here, the authors can utilize their “Random Seed” as the negative control for this comparison.

We thank the reviewer for these comments and the chance to add more discussion on these topics into the text. We also feel this study is a first step in establishing aptamers as an area ripe for more application of model-driven sequence optimization research.

As this was a pilot study, and based on AUC performance of the models, we felt an approach that tried to limit the distance from seed to novel predicted aptamer would be most likely to succeed in generating strong binders. As a small note, in our particular case any base pair change is permitted at any position along the aptamer. Therefore, we cannot walk the models until the model score is maximized globally as (theoretically) all sequences would converge to the same point. More generally, we were concerned that optimizing the model in this way was a potentially dangerous strategy. As indicated in Brookes et al., previous studies have shown that “many state-of-the-art predictive models suffer from pathological behaviour, especially in regimes far from the training data (Szegedy et al., 2014; Nguyen et al., 2015). Methods that optimize these predictive functions directly may be led astray into areas of state space where the predictions are unreliable and the corresponding suggested sequences are unrealistic” (Brookes et al., PMLR, 2019). We have added commentary in the discussion highlighting this important issue.

Additionally, although we acknowledge that our description may have been insufficient and/or confusing, we disagree with the assessment that the statement ““the ML directed

walks substantially outperform random walks and the original PD” does not have value. Our goal here was not to indicate the walking strategy was optimal. Rather, it was to indicate that the models had utility for pruning the search space. The comparison to random walks in different setting was meant to convey some degree of generality shown by the model as well highlight three different potential aspects/use-cases of the model, as indicated below:

- 1.) Starting from experimental seeds. By using sequences that were already good as a starting point it suggests that the model can be meaningfully used to improve candidates from experimental runs.
- 2.) Starting from random seeds. This was meant to convey that the models were not only able to locally improve sequences in which they already had tremendous context (e.g. the experimental seeds that they were partially trained on), but they had learned properties that could be applied to any sequence.
- 3.) Starting from ML-filtered candidates. This was meant to provide a putative mechanism for generating diverse novel candidates of reasonable score. Screening up to a billion candidates is still only a small part of a search space, so getting extremely high-quality candidates is unlikely. However, doing walks from such candidates provides a mechanism for diverse, potentially high-quality aptamers. This is distinct from simply performing extensive walks from random seeds which could lead the pathological behavior we described earlier.

Together these suggest that the models learned something somewhat general about what makes an aptamer good. We have adjusted the wording in the text to be more clear.

Indeed, it is for this reason that we feel like the paper already addresses this point by the reviewer *“The important & valuable key question is whether the CNN model is able to predict aptamer affinity accurately enough to significantly narrow down the experimental sequence space that needs to be tested.”*. The AUC numbers suggest that models have some signal above random in identifying potentially useful aptamers. However, we agree that AUC on a test set was insufficient, and thus we generated sequences (using the models) and experimentally validated that they perform well in practice. We see far more of these higher affinity aptamers (as a fraction of what we tested) than what was seen from random PD screening (especially on a per sequence basis). Moreover, we show this is not an artifact of the sequence starting points we chose by showing the relative enrichment compared to a random mutation process on the same sequence set. Together, we feel like this provides very strong evidence that the ML models are quite valuable for pruning the candidate aptamer search space.

10. *The conclusion “This demonstrates that MLPD can yield better aptamers than the examples it was trained with.” looks to be supported by the authors’ data, but the described comparisons between an “experimental seed” or a “ML seed” or a “randomly screened ML seed” don’t make sense. When considering the usage of the CNN predictions to extrapolate beyond the training set, the key question is how extrapolative accuracy depended on the training set sequences. For example, the CNN model’s predictions might be equally accurate on the entire aptamer*

sequence space or perhaps only a subset of the aptamer sequence space that is “closest” (by some metric) to the training sequence set. It would be worthwhile and interesting to quantify (e.g. using minimum edit or Hamming distance) the distance between the newly designed aptamer sequences and training set sequences and compare this distance to the CNN model accuracy. This is distinct from what is shown in Figure S5, which shows the edit distance between the designed sequence and the seed sequence.

We thank the reviewer for suggesting this analysis. We have now added an additional table which measures the AUC on the walked training set sequences grouped by distance from the initial seed. Here, we utilized the random walks since they were not already biased by the models (i.e. the models did not already think these were “good” aptamers). The table below shows the AUC at each stringency threshold and walked distance grouping (note, that higher stringencies were excluded since there were less than 10 points in any bin). There were a small number of “positive” points within any cell (14 to 87), so we also include the standard deviation of the AUC value based on performing 10,000 bootstraps.

model	Counts		Bin		SuperBin	
stringency	512nM	128nM	512nM	128nM	512nM	128nM
type	auc	auc	auc	auc	auc	auc
dist_range						
0-2	0.747 +/- 0.024	0.692 +/- 0.031	0.736 +/- 0.024	0.738 +/- 0.026	0.758 +/- 0.023	0.742 +/- 0.026
2-4	0.698 +/- 0.009	0.65 +/- 0.012	0.694 +/- 0.009	0.702 +/- 0.011	0.706 +/- 0.008	0.747 +/- 0.01
4-20	0.669 +/- 0.006	0.64 +/- 0.009	0.66 +/- 0.006	0.673 +/- 0.008	0.671 +/- 0.006	0.698 +/- 0.008

Notably, in cases where >2 stds changes were observed between rows (e.g. distance 4-20 compared to smaller distances), we were surprised to see that the model performance actually *improved* at greater distances. This is potentially a result of the models identifying that things far away from a training aptamer were less likely (on average) to be good (reinforcing the model’s utility at filtering out poor aptamers).

We have added the following text to the manuscript to summarize these findings:

“Notably, the model performance appeared to be independent of the distance to training set sequences. Model AUCs on randomly walked sequences from experimental seeds in the training set did not decline as the distance increased (Supp. Table ST7).”

11. *The authors don't provide any statistical analysis to support the importance of the GTGGATAG motif or indicate the location of this motif within the overall aptamer. There are always huge "edge effects" when randomly varying a sequence within a larger sequence region that could be biasing the motif importances. Unless the authors can provide more support for the importance of this motif, it should be removed from the text.*

We have updated the motif analysis to be more rigorous. We modified our strategy to use an existing tool for motif detection (MEME) which reports an e-value estimating the expected number of motifs with the given log likelihood ratio or higher. We employed this on the set of motifs predicted by the models and then verified that the motif was also enriched in the test data set. The shared top motif was "TGGATAG"; details are included below.

To assess the significance of the motif, we utilized MEME in differential enrichment mode to search for the top motifs in our ML and experimental data. First, we extracted the set of sequences walked from random seeds and compared these to the random seeds. The top motif (with e-value $3.2e-18$) is shown below.

We then examined the original particle display test sequences that were not used to train the model. Experimental test sequences observed in a positive pool were compared to the full set of experimental test sequences. The top motif (e-value = $3.3e-86$) is a close match to the model predicted motif.

These two analyses suggest a highly conserved 7-mer of *TGGATAG*.

We also examined the location of *TGGATAG* within the aptamers in the ML and observed positive data. While the magnitude of enrichment differs, the ML models and the observed data seemed to slightly prefer motifs starting near the very beginning (e.g. positions 1 and 2) and middle (between 10-16, 22).

Lastly, we examined the literature to see if this motif had been previously observed in aptamers to NGAL. A recent manuscript (Hong, et al. *J Transl Med.* 2019), identifies three high-affinity aptamers to NGAL. While the motif itself is not explicitly discussed in the manuscript, these three aptamers all contained the *TGGATAG* motif. The sequences

identified (excluding primers) are shown below with the motif highlighted. Notably it is the largest shared motif in the set of three sequences:

NA36: CCCATATGCTACTTTGCACACATCCT**TGGATAG**GGCT

NA42: CCGTGCGGATGTACAGGGACT**TGGATAG**TTTCTGA

NA53: GCGCT**TGGATAG**CAAGATCACGTTATCATCGTAAAC

We have added the following text to the manuscript to describe this:

“Sequence motifs are known to be important in aptamer affinity⁴². We first sought to identify the most frequent motif observed in aptamers selected by the ML models. To eliminate potential biases introduced by non-random seed sequences, we first examined differential enrichment between walked sequences and seed sequences in the ‘random seed’ set. Using MEME⁴³, we obtained a 7 nt motif (consensus motif = TGGATAG, e-value = 3.2e-18) shown in Supp. Fig. S4A. Next, we examined the original particle display test sequences that were not used to train the model. Experimental test sequences observed in a positive pool were compared to the full set of experimental test sequences. This yielded a highly similar 7nt motif (e-value = 3.3e-86) that shared the same consensus sequence, TGGATAG (Supp. Fig. S4B). Interestingly, while the motif was not explicitly discussed, a recent, independent study of high-affinity aptamers for NGAL included three distinct aptamer candidates also containing TGGATAG⁴⁴.”

12. The authors must carry out a structural similarity analysis to determine if all the high-affinity DNA aptamers fold into same/similar DNA structures. DNA structure plays a key role in creating the binding pockets for target binding. If the high affinity aptamers folded into similar structures, it would provide more evidence of their functionality. This analysis should be extended to the truncated DNA aptamers to determine if the truncated portion corresponds to a particular subset of the structure. As it stands, besides the Kd measurements, there is no physical/chemical analysis of the DNA aptamer sequences that explains WHY they bind to the target.

We agree with the reviewer and have updated the text to include secondary structure prediction results. Figure 4 has been updated to include secondary structure for both the full length and 23bp truncated aptamers. All 4 of these high-affinity aptamers fold into stem loop structures with the TGGATAG motif on the loop. The text has been modified to reference this figure and describe this result:

“Secondary structure prediction revealed that the 23nt truncated aptamer of G12 and G13 shared a hairpin structure with the TGGATAG motif located on the loop region (Fig. 4E, F). The same structure was observed in the full-length sequence for G12 and G13, suggesting the likely importance of this structure.”

Additionally, we have updated ST2 to include a note on whether this motif was observed in a loop for any of the top 10 ten lowest free energy structures (as returned by

ViennaRNA) and a description of our secondary structure analysis has been added to the methods under the subsection “Secondary Structure Analysis” (see below):

“Secondary Structure Analysis

DNA structures analysis was performed using the ViennaRNA package (v2.4.13)⁵⁵. Free energies for individual sequences were calculated at 37 °C using DNA parameters (Matthews model, 2004). The structure with the lowest free energy for the full-length and 23 nt truncation of G12, G13 were plotted in Fig. 4. To assess if the TGGATAG was observed in a loop, we visually inspected the top 10 lowest free energy predictions. Supp. Table ST3 identifies aptamers in which any of these predictions showed the motif completely contained within a hairpin.”

13. *When discussing the characterization of the truncated aptamers, the author should refer to the “seed” sequences as full-length sequences. The authors are using the word “seed” many times for different purposes and changing the definition.*

We have adopted the reviewer’s terminology, and replaced seed with “full-length sequence” when it refers to the initial aptamers used for truncation. We have also changed the columns headers in Supplemental Tables: ST3 (formerly ST2) header was changed from “Seed Origin” to “Sequence Origin” and the first header in ST8 (formerly ST6) had been “Seeds” but has been removed because it is not necessary.

14. *In the introduction, the authors need to explain what the NGAL target is and why it’s generally important/relevant as a biomarker.*

We have expanded the explanation of NGAL and its utility as a biomarker.

Reviewer #2:

This manuscript describes an machine-learning-guided “particle display” method to partition aptamers by affinity, and collects information to establish and train a “prediction model”. These predictions therefore support to identify high affinity motif and structure. The basic principle relies on the initial sequence and structure of parental aptamers which supports machine learning. The major finding of this study is to combine machine learning and physical validation to support the aptamer development – truncation, functional prediction, optimization, etc. Overall, Bashir et al. show an interesting study which provides a “shortcut” for aptamer development via intelligent sampling strategies. I can expect this approach would facilitate SELEX process, ad save time and energy in the aptamer development. However, I still have some concerns and find some limitation.

The major points

1) *PD measures the affinity of each aptamer with the target by fluorescence-based assay. How do the authors define the detection limitation (how sensitive it is)? How do the authors exclude non-specific binding? In addition to a protein as target of DNA aptamer selection, I am wondering if this system can be applied for live cell based SELEX? Or for RNA aptamer selection?*

We thank the reviewer for the questions, and respond to each in-line below:

How do the authors define the detection limitation (how sensitive it is)?

- The sensitivity is mainly limited by flow cytometer and the fluorescence of the detection reagent, with the limit of detection (LOD) = (Mean of the blank sample + 3 standard deviations). The fluorescence LOD for our reagent is ~600 RFU. The sensitivity of the K_D measurement will be determined by the fluorescence of the detection reagent. For example, the F_{max} for the antibody used in the study is ~15,000 RFU, so even 1/25 of the F_{max} can be distinguished from the background. The 1/3 F_{max} threshold used was well above the background fluorescence as compared to the blank sample.

How do the authors exclude non-specific binding?

- We agree that understanding the binding of an aptamer to its target compared to other non-target proteins is important in qualifying an aptamer for use in applications that involve complex media such as serum. However, in this manuscript, we focused on the affinity as a proof of concept to integrate particle display with machine learning and did not perform explicit screens to improve specificity. A future direction will be to implement ML with Multiparameter Particle Display to improve both affinity and specificity simultaneously ((2017, Multiparameter Particle Display (MPPD): A Quantitative Screening Method for the Discovery of Highly Specific Aptamers). We have updated the discussion to note this approach and include the relevant citation: "For example, to discover aptamers on the basis of both affinity *and* specificity ⁴, one can perform screens on a desired target and non-targets such as undesired homologues or protein mixtures, as we demonstrated in our previous work, Multi-Parameter Particle Display ²⁸."

In addition to a protein as target of DNA aptamer selection, I am wondering if this system can be applied for live cell based SELEX? Or for RNA aptamer selection?

- The PD method has been applied for RNA aptamer selection and chemically modified DNA aptamer selection. So the implementation of machine learning similar to this manuscript should be directly applicable. In principle, live cell based SELEX, or any other SELEX based approaches, can also take advantage of the methods described in the manuscript using ML models that extend the Counts model. However, the SELEX methods are based on competition, and the relative

enrichment (N) for two aptamers is limited by their relative K_D with $N = (K_{D1}+T)/(K_{D2}+T)$, where T is the free target concentration. So SELEX may take more rounds to generate count differences between two aptamers with close affinity (Detailed theoretical analysis can be found in 2014 Particle Display: A Quantitative Screening Method for Generating High-Affinity Aptamers). As the reviewer points out in their third question, counts are not always directly related to affinity. As a result, it will be interesting to see how well SELEX outputs can be used to train these ML models in the future.

2) *It was impressive to see only 2 rounds were effective to get good aptamers. How do the authors set up the "stringency criteria" and rationale?*

There are two parameters associated with "stringency criteria": fluorescence threshold and target concentration.

- We selected a fluorescence threshold of $F_{max}/3$ to maximize the accuracy of our affinity-based gating. As a threshold, it yields the maximum difference in fluorescence signal between an aptamer particle with a desired affinity and any aptamer particle that has higher (worse) K_D . For a detailed explanation, we have cited our previous work that provides the comprehensive theoretical analysis (2014 Particle Display: A Quantitative Screening Method for Generating High-Affinity Aptamers).
- We selected target concentrations empirically such that there was sufficient number of binders in the positive gate to be experimentally collected and used for ML model training.

3) *I am curious how the authors define the score and set up a sorting gate for the aptamer candidates in PD partitions? It seems the partition strategy really relies on the sequencing fraction – does it mean the frequency/abundance of sequences? If so, at this stage, no secondary structure is involved here? As our experience, in some cases a few sequences with low abundance also show good binding affinity due to their specific secondary structure. I am wondering how to avoid to filter out these potential sequences?*

Actually some papers described sequence-structure motif-based SELEX method. For example:

<https://pubmed.ncbi.nlm.nih.gov/27467247/>

AptaTRACE Elucidates RNA Sequence-Structure Motifs from Selection Trends in HT-SELEX Experiments

In this study, AptaTRACE can identify low-abundance motifs.

I am wondering if the study considers this point.

With regard to the reviewer's first question, the sorting gate was set at $F_{max}/3$ to maximize the accuracy of our affinity-based gating. The criteria of whether an aptamer

passed the threshold relies on the frequency at which the aptamer sequence appears in the positive gate compared to the negative gate.

The second set of issues the reviewer raises with regard to sequence abundance is an important one: the magnitude of sequencing counts are not always predictive of binding affinity. Indeed, this limitation may have caused the counts model to perform poorly in comparison to the Binned/SuperBin models in practice.

The binned models do not focus on sequencing counts but rather try to establish if a sequence was above or below the gating threshold. Low-abundance sequences should be similarly impacted in both the positive and negative pool, and we used a permissive bead fraction (0.2) to detect as many real sequences as possible. Though, as the reviewer indicated, very low abundant, but real, sequences could still be missed if that particular sequence has far fewer copies on a bead than expected.

In this work, we sought to minimize the inclusion of auxiliary information (e.g. secondary structure) information ML models, because we wanted to test if the ML models could identify high affinity aptamers in a completely data-driven manner. However, the AptTrace approach described by the reviewer could potentially be added to recover low-fraction aptamers. Specifically, if we were able to have such a strategy we could lower our stringency to recruit more positive aptamers, which could then improve the model. We have expanded the discussion to raise these two considerations (sequencing fraction and secondary structure). The updated sentences and corresponding citation are included below.

“In particular, models trained on particle display bins (with comparatively limited positive training examples) seemed to outperform the more fine-grained signal employed in the “Counts” model . . .

Additionally, we could extract more potential binders for model training from each sequencing round by using approaches that incorporate structure to identify even low-abundant aptamers with binding affinity⁵⁰.”

4) Although I am not familiar with the machine learning system, it was interesting to see the authors' principle to do it by using “seed sequences”. That does make sense to me. Please explain the rationale and whether any active motifs / binding core of aptamers will be selected in this stage?

Thank you for the comment. We handled identification of motifs and determination of binding cores as two distinct analyses in the text. Below, we discuss how we expanded on those two analyses in the revision.

Based on this comment, and a comment from Reviewer #1, we have clarified our use of the term seed sequences and added clarification to our motif analysis. Seed sequences simply serve as our starting point for ML-guided sequence perturbation. Here, active

motifs are observed only indirectly; as enriched subsequences in walked sequences when compared to the seeds. Our updated analysis (using the MEME tool) helped us identify the TGGATAG motif reported in results. Note that the model does not explicitly define these motifs. In contrast, the model tends to prefer sequences that contain motifs and in combination with other sequences features it identifies as important. Indeed, we believe this motif/structure agnostic, data-driven strategy is a strength of our approach, and demonstrates that the model enriched for high-affinity sequences without explicit, human-guided rules.

We identified binding cores via the truncation analysis. Again, here we attempted to have the model identify binding cores of aptamers by simply evaluating all possible subsequences (of particular lengths) and seeing which subsequences the model thought were highest scoring. However, some binding cores appear to be shared across our top scoring truncations. For example, all high-scoring truncations for G12 contain **ACGTTTTTGGTGGATAGCAAATG** and all high-affinity truncations for G13 contain **GAGGATTTGGTGGATAGTAAATC**. As with the motif analysis, we have added a more detailed description of these sequences in the results, reproduced below:

“In both cases, the high-performing 23-mer cores (**ACGTTTTTGGTGGATAGCAAATG** and **GAGGATTTGGTGGATAGTAAATC**) were contained in all larger truncations that reached the same K_D threshold suggesting these to be the key binding subsequences from the full-length sequence.”

Reviewers' Comments:

Reviewer #1:

Remarks to the Author:

The authors have responded well to this reviewer's comments.

Reviewer #2:

Remarks to the Author:

The authors have satisfactorily responded to the reviewer comments.